# Caffeine inhibits Notum activity by binding at the catalytic pocket

Yuguang Zhao [1]✉, Jingshan Ren [1], James Hillier[1], Weixian Lu[1] & Edith Yvonne Jones[1]✉

Notum inhibits Wnt signalling via enzymatic delipidation of Wnt ligands. Restoration of Wnt signalling by small molecule inhibition of Notum may be of therapeutic benefit in a number of pathologies including Alzheimer's disease. Here we report Notum activity can be inhibited by caffeine ($IC_{50}$ 19 µM), but not by demethylated caffeine metabolites: paraxanthine, theobromine and theophylline. Cellular luciferase assays show Notum-suppressed Wnt3a function can be restored by caffeine with an $EC_{50}$ of 46 µM. The dissociation constant ($K_d$) between Notum and caffeine is 85 µM as measured by surface plasmon resonance. High-resolution crystal structures of Notum complexes with caffeine and its minor metabolite theophylline show both compounds bind at the centre of the enzymatic pocket, overlapping the position of the natural substrate palmitoleic lipid, but using different binding modes. The structural information reported here may be of relevance for the design of more potent brain-accessible Notum inhibitors.

[1] Division of Structural Biology, Wellcome Centre for Human Genetics, University of Oxford, Oxford OX3 7BN, UK. ✉email: yuguang@strubi.ox.ac.uk; yvonne@strubi.ox.ac.uk

The Wnt signalling pathway is fundamental for embryonic development and adult tissue homoeostasis[1]. In the nervous system, Wnt signalling is important for neuronal differentiation, development[2] and neural stem cell maintenance[3]. Wnt ligands such as Wnt7a are essential for neural stem cell self-renewal and neural progenitor cell cycle progression[4]; Wnt5a can promote adult neuron stem cell differentiation and development[5]. FZD receptors, including FZD3, 5 and 10, are involved in neural development[6–8] and the Wnt co-receptor LRP6 is required for maintenance of neuron synapses[9].

Aging has been suggested to lead to downregulation of Wnt signalling and consequent attenuation of neurogenesis[10], potentially contributing to neurodegenerative diseases such as Alzheimer's disease[11]. A number of lines of evidence support this hypothesis. A common genetic variant (I1062V) of LRP6 reported to have slightly less Wnt signalling potency, is associated with late onset Alzheimer's diseases[12]. Higher levels of a Wnt negative modulator, Dickkopf-1, are associated with aging and involved in neuronal degeneration in Alzheimer's disease[13,14], while depletion of Dickkopf-1 restores neurogenesis in old-aged neural progenitor cells[15]. Collectively, these observations suggest that interventions targeted to increase Wnt signalling may offer therapeutic benefits in Alzheimer's disease.

We focused our attention on a recently characterized Wnt O-palmitoleoyl-L-serine hydrolase, Notum, which removes an essential Wnt lipid moiety to deactivate Wnt signalling[16]. Notum is important for neural development and head induction both in invertebrates (Planarian)[17] and vertebrates (Xenopus)[18]. Furthermore, Notum expression is detected in mammal brain hippocampal neurons[19]; Notum plays a key role in adult ventricular-subventricular zone (V-SVZ) neurogenesis[20]. Evidence exists that biological aging may result from increased Notum expression, while inhibition of Notum may rejuvenate stem cells[21]. Thus, small molecules that inhibit Notum could be used to restore impaired Wnt signalling and increase adult neurogenesis[20], which may be beneficial in pathologies such as Alzheimer's disease. A number of Notum inhibitors have been identified recently, however these molecules are unable to cross the blood-brain barrier[22–24]. To guide the development of brain-accessible Notum inhibitors, we turned our attention to the identification of natural products that can freely cross the blood-brain barrier. Our recent X-ray crystallography-based fragment screen identified a hit with similar chemical structure to melatonin, which led to the finding that this pineal gland secreted hormone can act as an inhibitor for Notum[25]. Caffeine and melatonin both show neuroprotective effects and caffeine can potentiate the effects of melatonin in the inhibition of Aβ oligomerization and modulation of the Tau-mediated pathway[26]. Additionally, caffeine is a chemical base of the Porcupine (Wnt palmitoleic lipid transferase) inhibitor,

ETC-159[27]. Both Porcupine and Notum act upon the same palmitoleic acid substrate. These clues prompted us to investigate whether caffeine could also target Notum. Here we demonstrate the ability of caffeine to bind to and inhibit Notum, properties that are not shared by caffeine metabolites. We also report the high-resolution crystal structure of the caffeine–Notum complex, which could serve as the basis for future attempts to develop more potent Notum inhibitory drugs.

## Results

**Caffeine inhibits Notum activity**. To investigate if caffeine could inhibit Notum activity, we first used an OPTS enzyme assay. OPTS is a common lipase substrate with an octanoyl lipid linked to a fluorescent moiety. It is not a natural Notum substrate; however, it can be used to quantify the enzymatic activity of purified Notum protein[23,25,28]. As shown in Fig. 1a, caffeine can inhibit Notum activity with an $IC_{50}$ of 19 μM. Caffeine (1,3,7-trimethylxanthine) is quickly metabolized by demethylation into paraxanthine (1,7-dimethylxanthine), theobromine (1,5-dimethylxanthine) and theophylline (1,3-dimethylxanthine)[29]. We could not detect obvious Notum inhibitory effects in the OPTS enzyme assay for any of the demethylated caffeine metabolites, although they only differ from caffeine by one methyl group. To further test if caffeine could inhibit Notum activity on its natural substrates, Wnt ligands, we employed a TOP-Flash assay to measure levels of Wnt/β-catenin signalling using a luciferase reporter[30]. Notum was mixed with caffeine at varying concentrations together with Wnt3a and applied to the HEK293-STF cell line (which harbours the TCF-LEF Super TOP-FLASH luciferase reporter[31]). Similar to a previous observation[18], we found that Notum did not completely suppress Wnt3a induced activity, (i.e. there is a baseline luciferase activity of ~30%), however, we observed that caffeine restored Notum-suppressed Wnt3a activity with an $EC_{50}$ value of 46 μM (Fig. 1b). To rule out any Notum-independent effects of caffeine on Wnt3a-dependent TCF/LEF activation, we performed a TOP-FLASH luciferase assay without the addition of Notum. Caffeine did not affect luciferase activity at concentrations up to 1 mM (Supplementary Fig. 1).

**Caffeine binds Notum directly**. To gain more direct, biophysical, evidence of caffeine binding to Notum, we carried out surface plasmon resonance (SPR) experiments. Notum was expressed, in vivo biotinylated in HEK 293T cells, and loaded onto a streptavidin chip as the ligand. Caffeine and its metabolites, paraxanthine, theobromine and theophylline were used as analytes. A clear dose-dependent response was recorded for caffeine (Fig.2a and Supplementary Fig. 2a). The calculated dissociation constant (Kd) is 85 μM. There is also weak binding by

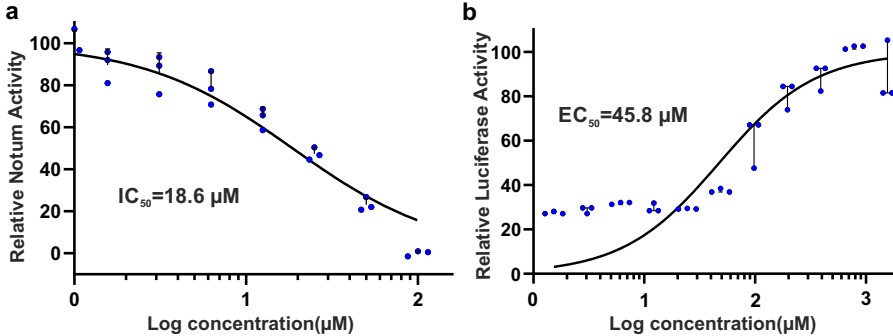

**Fig. 1 Notum inhibitory potency of caffeine. a** Notum in vitro enzyme assay with OPTS as substrate. **b** TOP-Flash luciferase assay measuring the effect of caffeine on the ability of Notum to suppress Wnt3a activity. The maximal response was set as 100%. The curves were fitted with nonlinear regression (log inhibitor concentration versus normalized response). The experiments were performed in triplicates ($n = 3$).

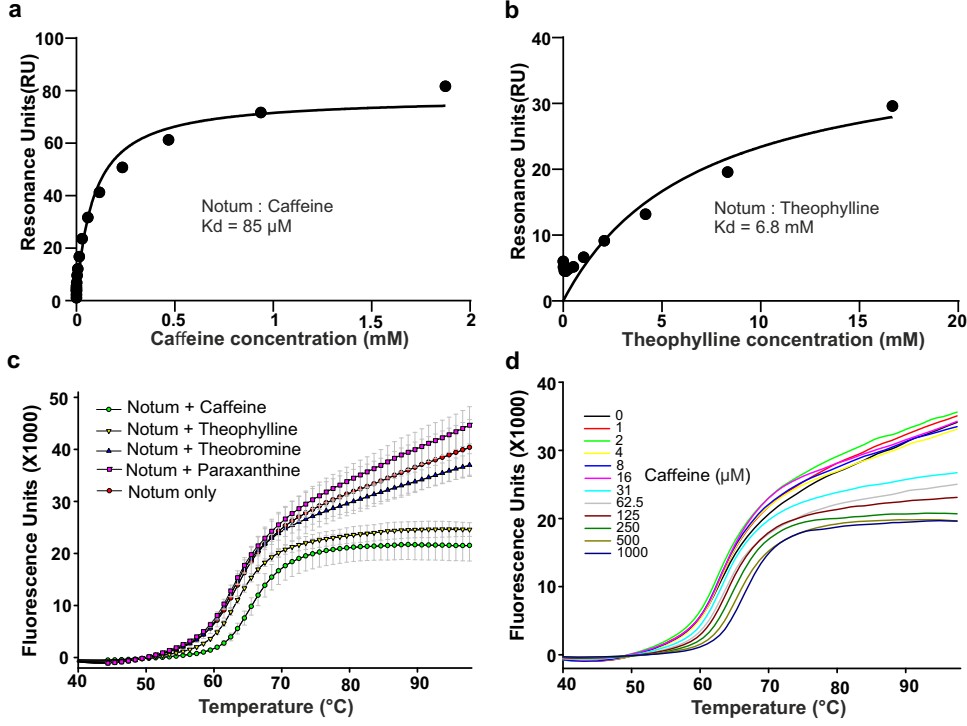

**Fig. 2 SPR and thermal shift assay analysis of caffeine and theophylline binding with Notum.** Biotinylated Notum was immobilized on a streptavidin chip. **a** Caffeine and **b** theophylline in 2-fold dilution series were used as analytes. **c** Thermal shift assay showing the melting curve of Notum with caffeine, theophylline and theobromine at concentrations of 500 μM. The experiment was performed in triplicates ($n = 3$). **d** Thermal shift assay showing the melting curve of Notum with caffeine in a 2-fold dilution series from 1–1000 μM.

theophylline (Fig. 2b and Supplementary Fig. 2b), with a calculated Kd value of 7 mM. We did not observe significant responses for paraxanthine or theobromine (Supplementary Fig. 2c, d).

To further confirm these interactions, we used a thermal shift assay to investigate caffeine and its metabolites binding to Notum. A Notum protein melting curve is shown in Fig. 2c. Notum protein in the absence of compounds has a melting temperature (Tm) of 63.6 °C, while in the presence of 500 μM caffeine, this was increased to 66.4 °C, a ΔTm of 2.8 °C. It is widely accepted that a ΔTm of 2 °C or more indicates a small molecule binding to a protein[32]. At the same concentration, theobromine (Tm 63.4 °C) and paraxanthine (Tm 63.5 °C) had little effect on the Tm, while theophylline (Tm 64.5 °C) conferred a ΔTm of about 1 °C, indicative of very weak binding to Notum. As shown in Fig. 2d, caffeine starts to increase the Tm of Notum from 32 μM in a dose-dependent manner.

**Caffeine binds at the enzymatic pocket of Notum.** To obtain structural evidence of caffeine and theophylline binding to Notum, we determined Notum complex structures using crystal soaking experiments as described in Methods. The data were collected at the Diamond Light Source and structures were determined by molecular replacement with our previously reported Notum structure as a search model (PDB code 6R8P)[23]. The caffeine and theophylline complex structures were refined at 1.53 Å and 1.24 Å, respectively, with good R factors and geometry (Table 1). Notum belongs to the α/β-hydrolase superfamily and its structure comprises a typical 8-stranded β-sheet flanked by α-helices (Fig. 3a). These β-strands and 3 α-helices (αB, αC and αF) form the α/β-hydrolase core, while the remaining α-helices (αA, αDs and αEs) form a mobile lid domain (Fig. 3a). Notum has a hydrophobic enzyme pocket (Fig. 3a). Both caffeine and

theophylline (Fig. 3b, c) bind in the Notum pocket and have well-defined electron density (Fig. 3d, e).

The overall Notum structures in both caffeine- and theophylline-bound complexes are similar to each other with an RMSD of 0.3 Å for 339 aligned Cα atoms. The structural superposition highlights a conformational change in the β8-αF loop, in which the side chains of residues H389, E390, and I391 in the theophylline-bound structure fold away from the enzyme pocket with a maximal mainchain movement of 4 Å (Fig. 3f). This conformation has not been observed in previously reported apo, inhibitor-bound or substrate-bound structures[16,22,23,25] and, as residue H389 is one of the catalytic triad (the other two residues being S232 and D340), β8-αF loop conformations are of interest for inhibitor design. However, the theophylline makes no interactions with the β8-αF loop. We note that the size of the crystal unit cell for the theophylline-bound structure is slightly bigger than that of the caffeine-bound structure (Table 1). It is possible this difference in lattice packing has allowed flexion of the Notum structure that is unrelated to the theophylline binding.

The Notum lid domain has been reported to adopt either an open or a closed conformation[25]. In comparison, the caffeine-bound structure reveals an intermediate conformation (Fig. 3g) as does the theophylline-bound structure. The loops around the movable lid domain are typically found to be flexible in Notum crystal structures with the corresponding regions of electron density often showing disorder. However, the αE1_αE2 loops are ordered in the caffeine-bound structure. Conversely, the β6_αD loop is partly disordered in the caffeine-bound structure, but ordered in previously reported open and closed structures (Fig. 3g).

**Caffeine–Notum interactions.** Notum has a well-defined hydrophobic pocket located between the α/β-hydrolase core and lid domains lined by the core domain residues W128 and Y129

**Table 1 Data collection and refinement statistics.**

|  | Caffeine_Notum | Theophylline_Notum |
|---|---|---|
| PDB code | 6TV4 | 6TUZ |
| Data collection beam line | DLS I24 | DLS I04-1 |
| Space group | $P2_12_12_1$ | $P2_12_12_1$ |
| Cell dimensions |  |  |
| $a$, $b$, $c$ (Å) | 60.0, 71.6, 78.3 | 60.4, 73.1, 78.9 |
| $\alpha$, $\beta$, $\gamma$ (°) | 90, 90, 90 | 90, 90, 90 |
| Resolution (Å) | 1.53 (1.56–1.53)[a] | 1.24 (1.26–1.24) |
| $R_{sym}$ or $R_{merge}$ | 0.084 (–) | 0.080 (–) |
| $I/\sigma I$ | 14.3 (1.5) | 13.1 (1.5) |
| Completeness (%) | 100 (98.4) | 100 (95.9) |
| Redundancy | 12.7 (10.6) | 11.4 (7.5) |
| Refinement |  |  |
| Resolution (Å) | 52.83–1.53 | 53.63–1.24 |
| No. reflections | 50884 (4944) | 98683 (9463) |
| $R_{work}/R_{free}$ | 0.170/0.214 | 0.201/0.212 |
| No. atoms |  |  |
| Protein | 2861 | 2899 |
| Ligand/ion | 34 | 148 |
| Water | 133 | 179 |
| B-factors |  |  |
| Protein | 25 | 20 |
| Ligand/ion | 50 | 41 |
| Water | 33 | 28 |
| R.m.s. deviations |  |  |
| Bond lengths (Å) | 0.005 | 0.010 |
| Bond angles (°) | 0.7 | 1.0 |

[a]Values in parentheses are for highest-resolution shell

from the αA_β2 loop; A233 and T236 from the αC helix; V187 from the β4_αB loop, and lid domain residues F268 from the β6_αD loop; P287, A290 and I291 from the αD helix; F319 and F320 from the αD'_ αD" loop; Q343 and V346 from the αE1 helix (Figs. 3 and 4). Caffeine, despite its polar nature, is bound within the hydrophobic pocket, and its interactions with Notum are predominately hydrophobic (Fig. 4a). It interacts with the majority of the pocket forming residues, including W128, V187, A233, T236, F268, P287, F319, and F320. The xanthine ring of caffeine forms ring stacking interactions with F268. Two oxygen atoms (O11 and O13, Fig. 3b) form hydrophobic interactions with P287 and T236. The three caffeine methyl groups (C10, C12, and C14, Fig. 3b) mediate hydrophobic interactions with F319 and F320; V187 and A233; and A342 and V346, respectively. These methyl group interactions may be important for caffeine binding, as we observed substantially impaired binding of demethylated caffeine metabolites to Notum. Paraxanthine (a major metabolite, missing the C12 methyl group from caffeine) and theobromine (a minor metabolite, missing the C10 methyl group from caffeine) do not bind Notum, while theophylline (another minor metabolite, missing the C14 methyl group from caffeine) maintains a very weak binding with a Kd value of 7 mM. Interestingly, we are still able to obtain a crystal structure of the Notum–theophylline complex. The theophylline binding mode is surprisingly different from caffeine. The theophylline xanthine ring is rotated ~180° relative to caffeine, but still forms ring stacking interactions with F268 (Fig. 4b, d). The oxygen atom O11 gains an interaction with Y129, which does not form contacts with caffeine. The oxygen O13 forms a contact with P287, which interacted with atom O11 of caffeine. The two methyl groups, C10 and C12, interact with W128, and V187 and T236, respectively.

When the caffeine-bound Notum structure is superimposed with the palmitoleic acid-bound structure (Fig. 4c), the caffeine molecule sits in the middle of the hydrophobic pocket, being centred on the position of the palmitoleic acid. The C14 methyl group of caffeine and the methyl group of palmitoleic acid occupy almost the same space (Fig. 4c), suggesting that this is a highly favourable position for methyl-binding. Theophylline lacks the C14 methyl group, however, the ~180° switch in its binding orientation within the pocket, compared to that of caffeine, positions the C10 methyl group of theophylline close to C14 methyl group of caffeine (Fig. 4d).

## Discussion

We report here that caffeine, a commonly consumed alkaloid, binds the Wnt deacylase Notum and inhibits its activity. In contrast, its metabolites paraxanthine and theobromine do not bind Notum; theophylline retains very weak binding, but is not sufficient to inhibit Notum enzyme activity. Caffeine only differs by one methyl group to each of its metabolites, however, all metabolites lost their Notum inhibitory effects. The overall Notum inhibitory potency of caffeine is relatively low ($IC_{50}$ 19 μM), although slightly greater than the reported potency of melatonin ($IC_{50}$ 75 μM)[25]. The natural Wnt substrate-based luciferase assay suggests an effective concentration of $EC_{50}$ 46 μM for restoring Notum inhibited Wnt signalling, while the biophysically determined affinity Kd is 85 μM.

For adults, consuming moderate amounts of coffee (3–4 cups a day, providing 300–400 mg of caffeine), the plasma caffeine concentration reaches approximately 15 μM[33] and its central nervous system (CNS) stimulating effects (through the adenosine receptors) are apparent at 2.5 μM[34]. Furthermore, caffeine is demethylated by cytochrome P450 isoform 1A2 in the human body[35] resulting in the major metabolite paraxanthine[36]. It is therefore unlikely that moderate coffee drinking would result in physiological Notum inhibition. However, caffeine is used as a therapeutic agent at much higher concentrations, for example in the treatment of apnea of prematurity where the plasma concentration can reach 150 μM[37]. At this concentration caffeine does not appear to have adverse effects on the brain[38] but would likely cause Notum inhibition.

Caffeine has been shown to have neuroprotective effects[39] and lowers the risk of Alzheimer's disease[40,41]. The mechanisms of these beneficiary effects are not well understood. One well-known possible mechanism is modulation of the adenosine A1 and A2A receptors[42,43], which requires micromolar physiological caffeine concentrations. Other mechanisms, such as the activation of ryanodine receptors and inhibition of phosphodiesterases require caffeine at a high, mM range, concentration[39]. Caffeine is also a weak acetylcholinesterase (AChE) inhibitor with an inhibitory constant (Ki) value of 175 μM[44]. Caffeine has been proposed as a lead compound for design and discovery of antagonists for monoamine oxidase B (MAO-B) and adenosine receptor A2A for the purposes of treatment of Parkinson's disease[45]. Our finding that caffeine (but not its metabolites) can inhibit Notum activity suggests there is an additional mechanism by which caffeine might exert neuroprotective effects, namely up-regulation of Wnt signalling. It is well documented that Wnt signalling can have neuroprotective effects and release of Wnt inhibition (such as genetic ablation of the Wnt inhibitor DKK1) can restore neurogenesis in old age[15]. Unlike DKK1, Notum is an enzyme with a druggable pocket and antagonism of Notum activity may be of therapeutic value for the treatment of neurodegenerative diseases. The Notum complex structures detailed here can inform rational small molecule design, which may ultimately lead to the development of clinically useful drugs.

In summary, we discovered that caffeine, but not its metabolites, binds to Notum and inhibits its enzyme activity. This observation may provide an additional mechanistic explanation,

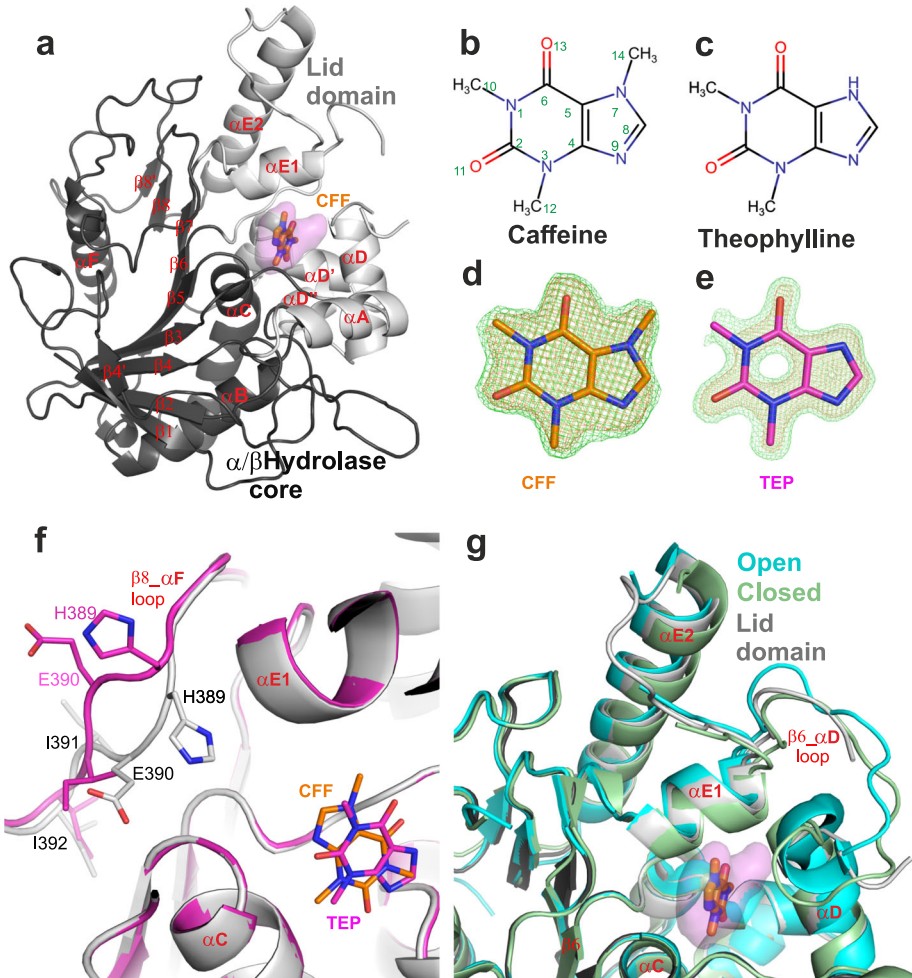

**Fig. 3 Overall structures of Notum complexes with caffeine and theophylline. a** Cartoon representation of Notum (grey) in complex with caffeine (CFF, PDB code 6TV4, brown sticks). The α/β-hydrolase core domain is in dark grey, the movable lid domain is in light grey. The enzyme pocket is shown as 80% transparent surface (pink). **b, c** Chemical structures of caffeine and theophylline. **d, e** Simulated annealing |Fo−Fc| omit electron density maps for caffeine (CFF, brown, PDB code 6TV4) and theophylline (TEP, magenta, PDB code 6TUZ). The maps were contoured at 3 σ and 5 σ, coloured in green and orange, respectively. **f** Close up view of caffeine- (grey cartoon and brown sticks) and theophylline- (magenta) bound structures with emphasis on β8_αF loop conformational changes. **g** Alignment of caffeine- bound structure with open (PDB code 4UYU, cyan) and closed (PDB code 4UZ1, pale-green) lid conformation structures.

increased Wnt signalling, for the neuroprotective role of caffeine. Our results suggest that caffeine intake from moderate coffee consumption may not cause Notum inhibition physiologically, but at therapeutic doses it may well do so. The high-resolution crystal structure of the caffeine–Notum complex may provide guidance for the design for more potent Notum inhibitory drugs.

## Methods

**Reagents**. Chemicals including caffeine, paraxanthine, theobromine, theophylline, biotin and the lipase substrate OPTS (8-octanoyloxypyrene-1,3,6-trisulfonate) are all from Sigma or Cayman chemical. Streptavidin (SA) sensor chips are from GE healthcare. Crystallization reagents are from Hampton Research.

**Protein production**. The human Notum enzyme core comprising amino acids S81–T451 with a C330S mutation[16] was cloned into a stable cell line vector pNeo_sec[46]. A stable polyclonal cell line was obtained by G418 selection of HEK293S GNTI- (ATCC CRL-3022) cells[47]. The stable cells were cultured in DMEM (high glucose, Sigma) supplemented with 1 mM glutamine, 1× non-essential amino acids and 10% foetal bovine serum (Invitrogen) at 37 °C with 5% CO₂. Large scale protein expression was performed by growing cells in expanded surface roller bottles (Greiner). For protein purification, the dialyzed conditioned media were passed through a 5 ml HisTrap Excel column (GE Healthcare) which was then washed with 20 mM imidazole PBS, and Notum protein was eluted with 300 mM imidazole PBS. To remove flexible glycans, the protein was deglycosylated

with endo-β-N-acetylglucosaminidase F1 (37 °C, 1 h) and further purified by size-exclusion chromatography (Superdex 200 16/600 column, GE Healthcare) in 10 mM Hepes, pH 7.4, 150 mM NaCl buffer.

**Notum activity assay**. The fluorescent lipase substrate OPTS was dissolved in water at a concentration of 20 μM. The enzyme assay mixtures were prepared with purified Notum protein at a concentration of 1 nM and compound (caffeine or its metabolites) at varies concentrations in buffer of 20 mM Hepes, pH7.4, 300 mM NaCl. The compound solutions with enzyme and OPTS solution were then mixed equally to a final volume of 100 μl in a 96 well flat bottom black polystyrol microplate and incubated at room temperature for 16 h. The fluorescence values were recorded using a Tecan Infinite F200 plate reader with an excitation wavelength of 435 nm (band-width 20 nm) and emission wavelength of 535 nm (bandwidth 25 nm). The inhibition curves were fitted with a one-site model with GraphPad Prism 8 software.

**Luciferase reporter assay**. The stable HEK293 STF cell line[31] carrying the Super Top Flash firefly luciferase reporter was split into 96-well plates and transfected 24 h later with a constitutive *Renilla* luciferase plasmid (pRL-TK, Promega) at 20 ng DNA per well using Lipofectamine 2000 (Life Technologies, UK). The media were replaced with test media 16 h later. The test media comprised conditioned medium from the mouse L-Wnt3a cell line (ATCC CRL-2647)[48], purified Notum (5 nM), and caffeine 2-fold dilution series from 6 nM to 1.6 mM concentration. The firefly and *Renilla* luciferase activities were measured 24 h later with Dual-Glo luciferase reporter assay system (Promega, Madison, WI) using an Ascent Lunimoskan luminometer (Lab-systems). The firefly luciferase activity was normalized to *Renilla* luciferase activity (relative light unit, RLU). Luciferase reporter assays were performed in triplicate.

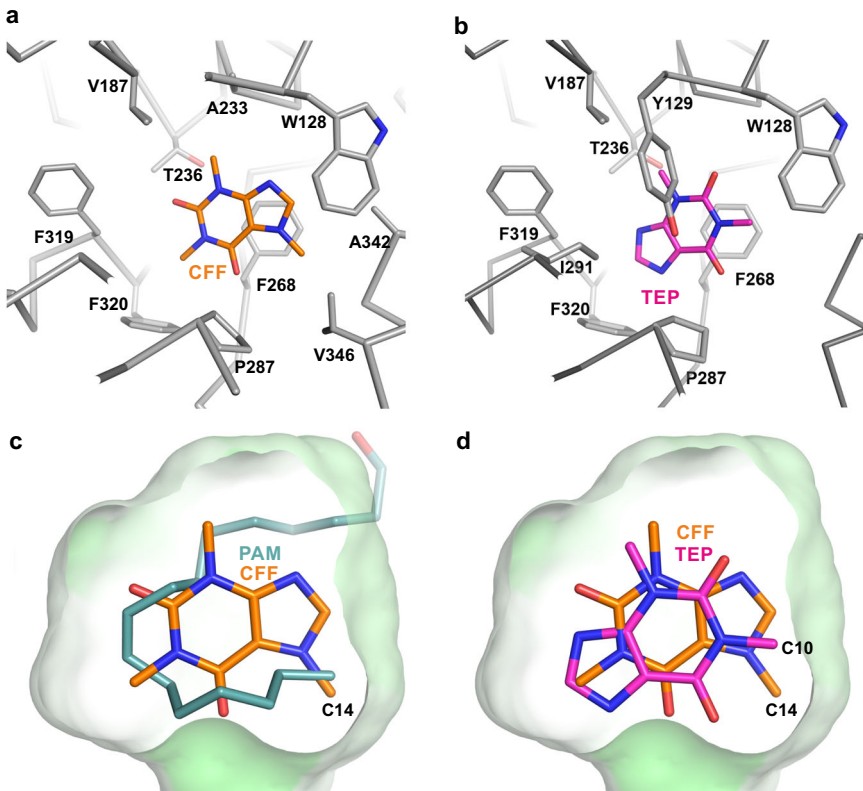

**Fig. 4 Details of Notum-inhibitor interactions. a** Notum-caffeine binding details (PDB code 6TV4). Notum (grey Cα trace) residues (grey sticks) interacting with caffeine (CFF, brown sticks) are defined with distance <3.9 Å. **b** Notum–theophylline (TEP, magenta sticks) binding details (PDB code 6TUZ). **c, d** The enzyme pocket is shown as surface (coloured green according hydrophobicity) with 50% transparency. The bound caffeine (brown) and theophylline (magenta) and palmitoleate (PAM, teal, PDB code 4UZQ) are shown as sticks.

**SPR equilibrium binding studies.** The human Notum core sequence was cloned in a pHL-Avi3 vector[49] and co-transfected with the pDisplay_BirA-ER plasmid[50] into HEK293T cells with the media supplemented with 20 μM biotin. This procedure allows *in vivo* biotinylation to occur[50], and the dialyzed conditioned medium was directly used for SA sensor chip immobilization without protein purification. The affinity was measured at 25 °C in 10 mM HEPES, pH 7.4, 150 mM NaCl, 0.005% Tween20, using a Biacore S200 machine (GE Healthcare). Approximately 2500 resonance units (RU) of the biotinylated Notum was coupled onto the chip. Caffeine or its metabolites in 2-fold serial dilution were used as analytes. The response was plotted versus the concentration of the analytes and fitted by nonlinear regression to a one-site saturation binding model (GraphPad Prism 8).

**Thermal shift assay.** Purified Notum protein (5 μg) in 25 μl assay buffer (10 mM Hepes, pH 7.4, 150 mM NaCl and 6× SYPRO Orange dye, Thermo Fisher Scientific, UK), was mixed with an equal volume of buffer containing either caffeine, or its metabolites: theophylline, paraxanthine, theobromine at 2-fold dilution from 1 mM to 1 μM. The samples were placed in a semi-skirted 96 well PCR plate (4-Titude, UK), sealed and heated in an Mx3005p qPCR machine (Stratagene, Agilent Technologies, USA) from room temperature at a rate of 1 °C/min for 74 cycles. Fluorescence changes were monitored with excitation and emission wavelengths at 492 and 610 nm, respectively. Reference wells, i.e. solutions without compounds were used to compare the melting temperature (Tm). The experiments were performed in triplicate.

**Crystallization and structure determination.** Crystallization screening was carried out using the sitting-drop vapour diffusion method[51] in 96-well Swissci/MRC plates. The crystallization drops contained 200 nl of Notum (5 mg/ml) and 100 nl of reservoir solution of 1.5 M ammonium sulfate and 0.1 M sodium citrate, pH 4.2. For crystal soaking, the caffeine or theophylline compound was mixed with crystal growing reservoir solution at a final concentration of 10 mg/mL and incubated for 20 min before harvesting. Soaked crystals were picked using a MiTeGen loop and cryo-protected by immersion into reservoir solution supplemented with 25% (v/v) glycerol or ethylene glycol, before being flash frozen in liquid nitrogen. Data sets were recorded from crystals at 100 K at the Diamond Light source (Didcot UK, beamline I04-1 and I24) with GDA(Generic Data Acquisition) software and processed with Xia2[52]. The structure was determined by molecular replacement with MOLREP[53] using our previously reported structure (PDB code 6R8P[23]) as a search

model. The model was then built with COOT[54] and finally refined with PHENIX[55]. The simulated omit map was generated by the CNS (Crystallography & NMR System) version 1.3[56]. The PyMOL Molecular Graphics System (Schrödinger, LLC.) was used to prepare figures.

**Reporting summary.** Further information on research design is available in the Nature Research Reporting Summary linked to this article.

## Data availability

Coordinates and structure factors have been deposited in the Protein Data Bank under accession code 6TV4 and 6TUZ for the Notum complex with caffeine and theophylline respectively. All relevant data are available from Y.Z. upon request. All source data underlying the graphs and charts presented in the main figures and supplementary figures are available as Supplementary Data 1–10.

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

## Acknowledgements

The authors thank the Diamond Light Source I04-1 and I24 beam line for assistance of data collection (under BAG application MX14744). We thank Dr Matthias Zebisch (Evotec), Prof. Jean-Paul Vincent (Crick Institute), Prof. Paul Fish (Alzheimer's Research UK UCL Drug Discovery Institute) and Patricia Salinas (University College London) for their helpful discussion. We thank Drs Laura Diaz Saez and Oleg Fedorov (TDI, Oxford) for assistance with using the Biacore S200 machine. This work was funded by Cancer Research UK, the UK Medical Research Council (to E.Y.J., C375/A17721 and MR/M000141/1) and the Wellcome Trust (grant 203141/Z/16/Z supporting the Wellcome Centre for Human Genetics). This is a contribution from the UK Instruct-ERIC Centre.

## Author contributions

Y.Z. and E.Y.J. designed the project and wrote the manuscript together with J.R. and J.H. Y.Z. performed experiments and analyzed data with J.R. and J.H. W.L. performed mammalian expression.

## Competing interests

The authors declare no competing interests.
