## [Peer Review File · Communications Biology]

Reviewers' comments:

Reviewer #1 (Remarks to the Author):

I am a Wnt biologist, not a structural biologist. Notum is an interesting enzyme in the Wnt pathway, de-acylating Wnts to end their signaling function. Several structures of Notum in complex with palmitate and various small molecules have already been published. Here the authors investigate the interaction of caffeine and related small molecules on the activity of Notum, and solve the crystal structures of Notum complexed to caffeine and theophylline. The studies appear to be well done, and the biochemistry is solid. The findings are of interest to those who drink coffee, get older, and/or think about inhibitors of Notum. I have no suggestions for further enhancements to this solid in vitro study.

Reviewer #2 (Remarks to the Author):

Zhao et al. present a novel mechanism of caffeine-mediated restoration of Wnt signalling through inhibition of the palmitoleoyl-protein carboxyl esterase Notum. The manuscript provides original and novel insights into how caffeine acts as an inhibitor for Notum, indicating its potential as an activator of Wnt signalling. Furthermore, the presented findings offer a starting point for the rational design of small molecules targeting the Notum-Wnt interaction that may provide a therapeutic intervention strategy for diseases associated with decreased Wnt signaling, such as Alzheimer's disease.

The presented claims and models are supported by biophysical and high-resolution crystallography experiments that reveal the binding modes of both caffeine and theophylline to the Notum catalytic pocket. Detailed insights into the conformational changes of Notum induced by caffeine or theophylline binding are shown. The authors use their findings to convincingly explain the observed differences in the IC₅₀, EC₅₀ and K_d values of caffeine and theophylline that were determined in Notum activity assays, TCF/LEF luciferase reporter assays and SPR analysis for Notum binding.

Overall, the presented work is of high quality and provides novel and valuable insights that hold relevance not only for the Wnt field, but also for aging-related diseases. There are a few concerns that need to be addressed to strengthen the manuscript:

1. In Fig. 1b, HEK293-STF cells are supplied with Notum, Wnt3A and caffeine. Here, a control experiment should be added in which HEK293-STF cells are supplied only with caffeine and Wnt3a (without Notum). This control would be required to rule out Notum-independent effects of caffeine on Wnt3a-dependent TCF/LEF activation.
2. As mentioned in the introduction, caffeine also served as a chemical base for development of the Porcupine inhibitor ETC-159. As Notum and Porcupine are natural antagonists, a potential inhibitory effect of caffeine on Porcupine will interfere with the Wnt activating effect aimed for in this manuscript. What are the effects of caffeine on Wnt secretion at relevant concentrations?
3. There is very little background provided on Notum expression in the brain, and its link to Alzheimer's appears rather indirect. Where and when is Notum expressed in adult brains, and by which cells? Is Notum expression increased with age? A more elaborate description of the role of Notum in the brain would help the reader to place the current findings in context.

Reviewer #3 (Remarks to the Author):

The Wnt deacylase Notum is a potential drug target for Alzheimer's and other aging diseases. In this manuscript, combining in vitro activity and binding assays with structural analysis, Zhao et al. demonstrated that caffeine is a natural inhibitor of Notum. Overall, this manuscript is well-written, with a solid and interesting story.

Specific issues:

1. In Fig. 1a, the standard error for IC50 should be included.
2. Fig 2, raw SPR binding data for caffeine and its metabolites, paraxanthine, theobromine and theophylline should be included in this manuscript (at least as supplementary material).
3. To strengthen the direct binding conclusion, an additional assay (especially ITC) is strongly encouraged.
4. With a Kd of 7mM, the electron density for TEP (Fig 3e) is amazingly good and supports the assigned binding mode. To better demonstrate the binding mode, with such a high resolution, authors may want to show the CFF density using a contour level higher than 3σ (Fig 3d).

Reviewer #1 (Remarks to the Author):

I am a Wnt biologist, not a structural biologist. Notum is an interesting enzyme in the Wnt pathway, de-acylating Wnts to end their signaling function. Several structures of Notum in complex with palmitate and various small molecules have already been published. Here the authors investigate the interaction of caffeine and related small molecules on the activity of Notum, and solve the crystal structures of Notum complexed to caffeine and theophylline. The studies appear to be well done, and the biochemistry is solid. The findings are of interest to those who drink coffee, get older, and/or think about inhibitors of Notum. I have no suggestions for further enhancements to this solid in vitro study.

Response: We thank the reviewer for the encouraging comments.

Reviewer #2 (Remarks to the Author):

Zhao et al. present a novel mechanism of caffeine-mediated restoration of Wnt signalling through inhibition of the palmitoleoyl-protein carboxyl esterase Notum. The manuscript provides original and novel insights into how caffeine acts as an inhibitor for Notum, indicating its potential as an activator of Wnt signalling. Furthermore, the presented findings offer a starting point for the rational design of small molecules targeting the Notum-Wnt interaction that may provide a therapeutic intervention strategy for diseases associated with decreased Wnt signaling, such as Alzheimer's disease.

The presented claims and models are supported by biophysical and high-resolution crystallography experiments that reveal the binding modes of both caffeine and theophylline to the Notum catalytic pocket. Detailed insights into the conformational changes of Notum induced by caffeine or theophylline binding are shown. The authors use their findings to convincingly explain the observed differences in the IC₅₀, EC₅₀ and K_d values of caffeine and theophylline that were determined in Notum activity assays, TCF/LEF luciferase reporter assays and SPR analysis for Notum binding.

Overall, the presented work is of high quality and provides novel and valuable insights that hold relevance not only for the Wnt field, but also for aging-related diseases. There are a few concerns that need to be addressed to strengthen the manuscript:

Response: We thank the reviewer for the encouraging comments and constructive suggestions.

1. In Fig. 1b, HEK293-STF cells are supplied with Notum, Wnt3A and caffeine. Here, a control experiment should be added in which HEK293-STF cells are supplied only with caffeine and Wnt3a (without Notum). This control would be required to rule out Notum-independent effects of caffeine on Wnt3a-dependent TCF/LEF activation.

Response: We have added the control experiment of luciferase assay without added Notum protein, shown in supplementary Figure 1. The results show caffeine does not influence Wnt3a-induced TOP-Flash luciferase activity. The text is modified at page4 Line8-11: "To rule out any Notum-independent effects of caffeine on Wnt3a-dependent TCF/LEF activation, we performed a TOP-FLASH luciferase assay without the

addition of Notum. Caffeine did not affect luciferase activity at concentrations up to 1 mM (supplementary Fig.1)”.

2. As mentioned in the introduction, caffeine also served as a chemical base for development of the Porcupine inhibitor ETC-159. As Notum and Porcupine are natural antagonists, a potential inhibitory effect of caffeine on Porcupine will interfere with the Wnt activating effect aimed for in this manuscript. What are the effects of caffeine on Wnt secretion at relevant concentrations?

Response: We thank the reviewer for the intriguing question. A small chemical change in a small molecule can make a big difference in biological effects, as we see in the case of theobromine: losing only one methyl group results in a total loss of Notum inhibitory effects. ETC-159 contains the caffeine base, however the base only accounts for about half of its structure. It therefore appears less likely ETC-159 and caffeine would have similar effects on a target. We do not know caffeine's effects on Wnt secretion. It would be interesting to know and this warrants further investigation, however it is beyond the scope of this report.

3. There is very little background provided on Notum expression in the brain, and its link to Alzheimer's appears rather indirect. Where and when is Notum expressed in adult brains, and by which cells? Is Notum expression increased with age? A more elaborate description of the role of Notum in the brain would help the reader to place the current findings in context.

Response: We thank the reviewer for this nice point. Notum was only identified as a Wnt deacylase relatively recently (it is 5 years since the first report, ref# 16) and there are a lot of interesting questions to be investigated. There is no direct report of Notum expression in healthy human brain or Alzheimer's brain, however, there are rat hippocampal neuron RNAseq data that support Notum brain expression (ref#19), and Notum effects on the planarian and Xenopus brain are well known. Our collaborators in an Alzheimer's Research UK supported consortium are actively pursuing this line of research. We have modified the text in page2 Line21-25: “Notum is important for neural development and head induction both in invertebrates¹⁷ (Planarian) and vertebrates¹⁸ (Xenopus). Furthermore, Notum expression is detected in mammal brain hippocampal neurons¹⁹. While the roles of Notum in human brain function, aging and neuronal degeneration remain to be investigated”.

Reviewer #3 (Remarks to the Author):

The Wnt deacylase Notum is a potential drug target for Alzheimer's and other aging diseases. In this manuscript, combining in vitro activity and binding assays with structural analysis, Zhao et al. demonstrated that caffeine is a natural inhibitor of Notum. Overall, this manuscript is well-written, with a solid and interesting story.

Response: We thank the reviewer for the positive comments and valuable suggestions.

Specific issues:

1. In Fig. 1a, the standard error for IC50 should be included.

Response: We have updated Fig.1a, it now includes standard error bars.

2. Fig 2, raw SPR binding data for caffeine and its metabolites, paraxanthine, theobromine and theophylline should be included in this manuscript (at least as supplementary material).

Response: We have included the raw SPR binding data for caffeine and its metabolites, paraxanthine, theobromine and theophylline in supplementary Fig. 2. They are also cited in the text: page 5 line4-7.

3. To strengthen the direct binding conclusion, an additional assay (especially ITC) is strongly encouraged.

Response: We appreciate the reviewer's suggestion. As our laboratory currently does not have access to ITC equipment, we have added results from a thermal shift assay (also known as differential scanning fluorimetry) which is also frequently used for characterization of small molecule-protein interactions. Two panels are added in Fig.2, with method added in Page16 Line7-16. The Figure legend has been modified in page 6 Line2-7. And main text has been modified in Page5 Line 8-17.

4. With a K_d of 7mM, the electron density for TEP (Fig 3e) is amazingly good and supports the assigned binding mode. To better demonstrate the binding mode, with such a high resolution, authors may want to show the CFF density using a contour level higher than 3σ (Fig 3d).

Response: We have update Fig 3d and 3e, with two contour levels, 3σ and 5σ , coloured in green and orange, respectively. The Figure legend has been modified in page 8 Line 10.